# Sex-specific exploration accounts for differences in valence learning in male and female mice

Heike Schuler[1], Eshaan S Iyer[1†], Gabrielle Siemonsmeier[1†], Ariel Mandel Weinbaum[2], Peter Vitaro[2], Shiqing Shen[2], Rosemary C Bagot[2,3]*

[1]Integrated Program in Neuroscience, McGill University, Montreal, Canada; [2]Department of Psychology, McGill University, Montreal, Canada; [3]Ludmer Centre for Neuroinformatics and Mental Health, Montreal, Canada

**Abstract** Valence, the quality by which something is perceived as good or bad, appetitive or aversive, is a fundamental building block of emotional experience and a primary driver of adaptive behavior. Pavlovian fear and reward learning paradigms are widely used in preclinical research to probe mechanisms of valence learning but with limited consideration of sex as a biological variable despite known sex differences in neuropsychiatric disorders associated with impaired valence. Here, we compare appetitive-only, aversive-only, and mixed-valence cue–outcome Pavlovian conditioning paradigms in male and female mice to dissociate effects of context, valence, and salience in a sex-specific manner. Using a data-driven approach to identify behaviors indicative of valence learning in an unbiased manner, we compare task performance between paradigms in male and female mice. We show that while male and female mice acquire appetitive and aversive associations in both single- and mixed-valence paradigms, sex differences emerge in single-valence paradigms. Ultimately, we show that these apparent sex differences in valence learning are driven by non-specific baseline differences in exploratory behavior. Males explore more at baseline, altering their trajectory of cue–reward association acquisition, whereas females explore less at baseline, increasing shock-facilitated freezing in aversive-only contexts, masking cue discrimination. Overall, our findings illustrate how task design differentially impacts behavioral expression in male and female mice and demonstrate that mixed-valence paradigms afford a more accurate assessment of valence learning in both sexes.

## Editor's evaluation

This important manuscript provides a comprehensive analysis of male and female mice across aversive, appetitive, and mixed-valence Pavlovian conditioning. Using pose estimation, the authors show that standard behavioural measures capture only a fraction of the substantial behavioural reorganization that occurs. They present convincing evidence that animals face competing demands to either respond to cues or explore the environment but cannot do both simultaneously. These findings will be of interest to behavioural neuroscientists and learning theorists studying sex differences.

## Introduction

Understanding how emotion is represented in the brain remains a major challenge for modern neuroscience. The complex and subjective nature of human emotional experience challenges efforts to uncover its mechanism. While emotion is complex, valence, the intrinsic attractiveness or aversiveness of a stimulus or state, is fundamental and translates across species (*Adolphs et al., 2019*; *Anderson*

*For correspondence:
rosemary.bagot@mcgill.ca

†These authors contributed equally to this work

Competing interest: The authors declare that no competing interests exist.

**eLife digest** Emotions are fundamental to our experience and essential to our survival. They shape how we react to the world, interact with others or make decisions. But emotions are difficult to study because they are complex and subjective. One key building block of emotion is valence – the sentiment that something feels good (positive) or bad (negative).

Valence strongly shapes behavior. We avoid unpleasant experiences and seek out rewarding ones. Animals, including humans, learn to anticipate such outcomes when neutral cues become linked to positive or negative events, a process called associative learning. Problems in this type of learning are central to many psychiatric disorders. For example, anxiety can involve overgeneralizing fear, depression often shows reduced positive memories with a bias toward negative memories, and post-traumatic stress disorder is marked by intrusive traumatic associations. These disorders also show clear differences between men and women.

To study the biology of valence learning, researchers often use mice and train them to associate cues with either rewards, i.e., 'appetitive outcomes', or punishments, i.e., 'aversive outcomes'. However, most studies look at only one type of outcome in isolation, even though real-life situations often involve both. In addition, research has often included only male mice or applies behavioral phenotypes established in male mice to females without considering the influence of sex differences.

Schuler et al. set out to answer how the presence of only positive, or only negative or both valences together influences associative learning and its behavioral expression in male and female mice. This question arose because few studies have examined mixed-valence learning, and findings on sex differences in reward and threat learning are inconsistent. We need to understand this because sex differences are prevalent in valence-related psychopathologies, and we need robust models to study these phenomena in both sexes.

The researchers developed a novel mixed-valence conditioning protocol, in which mice are exposed to both positively and negatively valenced stimuli in the same context. They then identified predictors of learning in a data-driven way and found that field-standard metrics perform poorly in evaluating performance. Applying data-driven metrics to evaluate performance during training revealed sex-specific behaviors in the single-valenced protocols; for example, female performance suggested poor discrimination of cues in the aversive-only protocol, while males appeared to take longer to acquire positive associations. These apparent sex differences were not observed in mice trained in the mixed valence protocol. This shows that male and female mice can learn equally well about appetitive and aversive outcomes.

These apparent sex differences in learning are due to an underlying sex difference in exploration, where males have a higher baseline level of exploratory behaviors. Finally, they demonstrated that females are more sensitive to the delivery of repeated foot shocks, showing higher generalization of pausing response in the aversive-only protocol. This difference is not indicative of a lack of learning, but rather a consequence of differences in exploration specific to the aversive-only context.

Schuler et al. present a novel mixed-valence conditioning protocol for freely moving mice. This protocol overcomes limitations in classically employed fear- and reward-conditioning protocols and will allow researchers in behavioral and systems neuroscience to disentangle behavioral and neural correlates of normative salience and valence processing, as well as disruptions in animal models for psychiatric disorders. This work highlights the critical importance of considering sex when establishing behavior protocols and describes a novel approach for rigorously evaluating in-depth behavioral phenotyping data, a fundamental concern that has received far less attention than the development of phenotyping tools themselves.

*and Adolphs, 2014*; *Gündem et al., 2022*). Valence guides adaptive behavior: aversive stimuli are avoided, and appetitive stimuli elicit approach. Through associative learning, cues that predict valenced stimuli shape behavior in anticipation of the outcome. Disruptions in valence learning are implicated in a variety of neuropsychiatric disorders (*Admon and Pizzagalli, 2015*; *Di Chiara, 1999*; *Lissek et al., 2005*). For example, patients with major depressive disorder display negative biases in memory recall (*Brittlebank et al., 1993*; *Douglas and Porter, 2010*; *Elliott et al., 1997*; *Maniglio et al., 2014*; *Noworyta et al., 2021*), anxiety disorders are generally defined by maladaptive threat

generalization (*Lee et al., 2024*; *Lissek et al., 2014*; *Vandael et al., 2025*), and patients with post-traumatic stress disorder suffer from intrusive memories of traumatic events (*Ehlers and Clark, 2000*). Notably, sex differences exist in the prevalence, symptom presentation, and severity of these disorders (*Kuehner, 2003*; *McLean et al., 2011*).

Preclinical Pavlovian conditioning protocols are widely used to study mechanisms of valence learning in rodents. Mice or rats learn to associate either an auditory or visual cue or a context (conditioned stimulus; CS) with an inherently valenced stimulus, such as an electric foot shock (unconditioned stimulus; US). Fear conditioning is widely used as a model of associative valence learning. Reward conditioning protocols are less frequently employed, partially because rodents take substantially longer to form reward than fear associations (*Deseyve et al., 2024*; *Domingues et al., 2025*). In both instances, animals learn about a single valence in isolation, yet real-world environments commonly engender outcomes of both valences. The presence of opposite valence outcomes is known to influence learning, yet this has received little attention of late (*Konorski, 1973*; *Laurent et al., 2022*). Single-valence tasks can conflate cue and context learning and confound valence with salience because valenced outcomes are inherently salient. Despite this, limited research has simultaneously examined appetitive and aversive learning in part due to the scarcity of protocols that dissociate opposing valence processes, especially in mice (*Lefner and Moghaddam, 2024*; *Ray et al., 2022*; *Ray et al., 2020*).

To date, few studies have examined associative learning in mixed-valence contexts and even fewer have examined both sexes. The few studies that have considered sex as a biological variable point to potential sex differences in both threat and reward learning (*Borkar et al., 2020*; *Keiser et al., 2017*; *Lefner et al., 2022*). However, findings are inconsistent and limitations in experimental designs preclude a clear understanding of the interaction of sex and valence learning. This is particularly important given prevalent sex differences in valence-related psychopathologies, in which often processing of both positive and negative valences is altered (*Kuehner, 2003*; *McLean et al., 2011*). Here, we asked how the presence of more than one valence influences learning and behavioral expression of learning in male and female mice to comprehensively profile sex differences in valence learning. We trained male and female mice in purely appetitive, purely aversive, or mixed-valence conditioning tasks and used data-driven behavioral profiling combined with a partial least squares discriminant analysis to distill learning-relevant behaviors in both sexes. We show that both sexes acquire appetitive and aversive associations, yet task design differentially influences behavioral expression of learning-relevant behaviors in males and females. Examining the evolution of learning across tasks revealed baseline sex differences in exploratory behaviors that mask learned cue discrimination in a context-dependent manner. Ultimately, we show that a mixed-valence context minimizes sex differences in exploration to more accurately probe valence learning in both sexes.

## Results

### Standard metrics of valence learning suggest poor task performance

Because valenced stimuli are inherently salient, single-valence paradigms conflate salience and valence. To better isolate valence encoding, we developed a paradigm in which mice learn appetitive and aversive associations in parallel. In this mixed-valence paradigm, one auditory stimulus was paired with chocolate milk ($CS^R$), another auditory stimulus was paired with mild footshock ($CS^S$), and a third auditory stimulus was non-reinforced ($CS^-$) (*Figure 1A*). We compared the behavior of male and female mice trained in the mixed paradigm to mice trained in either an appetitive-only paradigm (only $CS^R$ and $CS^-$), or an aversive-only paradigm (only $CS^S$ and $CS^-$).

We first quantified performance in all three conditioning paradigms using field standard metrics: number of head entries to evaluate appetitive learning and percent time freezing to evaluate aversive learning. Quantifying behavioral responding during the first 10 s of each cue (prior to outcome delivery) suggested limited evidence of appetitive acquisition in the mixed or appetitive-only paradigm, with significant increases in head entries to the $CS^R$ only at training end in female (*Figure 1B*) or male (*Figure 1C*) mice. This is consistent across other food port metrics (*Figure 1—figure supplement 1*). In contrast, we observed stronger evidence of $CS^S$ acquisition, yet behavioral responding varied between males and females across paradigms (*Figure 1D, E*). In the mixed paradigm, both males and females freeze more to the $CS^S$ than either $CS^-$ or $CS^R$, whereas in the aversive-only paradigm,

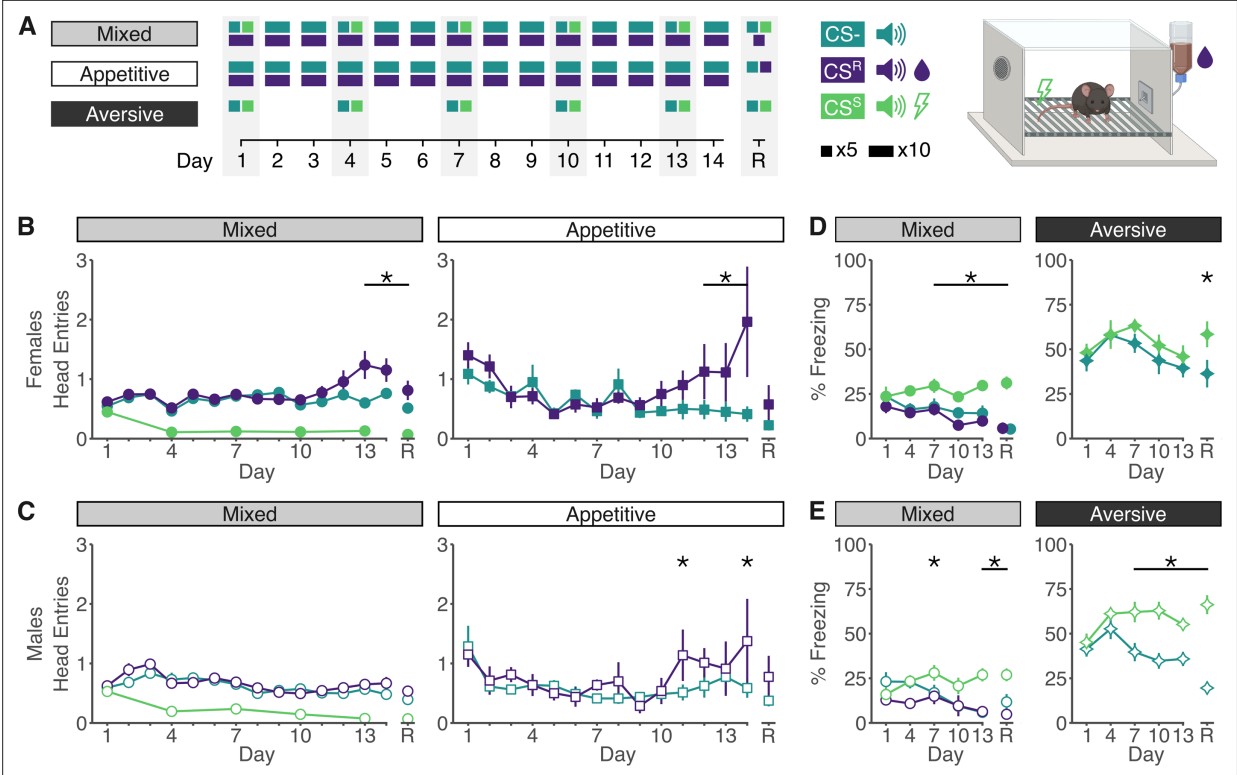

**Figure 1.** Standard indicators of valence learning point to poor task performance. (**A**) Overview of the experimental design and apparatus. (**B**) In females, head entries in the first 10 s of the cue are significantly higher to the CS$^R$ compared to the CS$^-$ at the end of training in both mixed (left) and appetitive-only (right) paradigms. On recall day (R), discrimination in head entries only remains significant in the mixed paradigm. (**C**) Male mice are performing more head entries to the CS$^R$ compared to the CS$^-$ only at the end of the appetitive-only paradigm, which returns to non-significance on recall day. ***Supplementary file 1a*** contains full statistical analyses for both sexes, including comparison to CS$^S$. (**D**) In females, percent time spent freezing is increased in response to CS$^S$ compared to CS$^-$ in the mixed-valence paradigm, but not in the aversive only paradigm throughout training. On recall day, females have significantly higher freezing levels to the CS$^S$ than the CS$^-$ in both paradigms. (**E**) Male mice have increased CS$^S$ freezing in the mixed and aversive-only paradigm as compared to CS$^-$, and discrimination remains significant in both paradigms on recall day. ***Supplementary file 1b*** contains full statistical analyses for both sexes, including comparison to CS$^R$. *$p_{adj} < 0.05$. Errorbars indicate SE. Sample sizes: Mixed: $n_{Male} = 37$, $n_{Female} = 37$; Appetitive: $n_{Male} = 8$, $n_{Female} = 8$; Aversive: $n_{Male} = 8$, $n_{Female} = 8$.

The online version of this article includes the following figure supplement(s) for figure 1:

**Figure supplement 1.** Port-based readouts of reward learning.

**Figure supplement 2.** % Freezing quantified using different motion threshold and freezing duration parameters in ezTrack.

males again freeze more to the CS$^S$ than CS$^-$, but females show high levels of freezing to both cues. These observations are robust across a range of freezing thresholds (***Figure 1—figure supplement 2***). Further, overall freezing levels were higher in the aversive-only paradigm. Due to the lack of evidence for reward learning, as well as the paradigm differences in freezing, we asked if these standard behavioral metrics accurately capture behavioral expression of learning.

## Data-driven analysis identifies robust behavioral predictors of cue identity

Standard behavioral metrics quantify a predetermined subset of all behavioral variability. In contrast, data-driven methods enable comprehensive profiling of the full behavioral repertoire. Given increasing recognition that task parameters and sex can influence behavioral expression, we applied DeepLabCut (DLC; *Lauer et al., 2022*; *Mathis et al., 2018*; *Nath et al., 2019*) and Keypoint-MoSeq (*Weinreb et al., 2024*) to agnostically identify and group recurrent behavioral patterns (***Figure 2—figure supplement 1***; ***Supplementary file 1c***). We reasoned that if mice had indeed acquired cue–outcome associations, some aspect of behavioral patterns would predict cue type (CS$^S$, CS$^R$, and CS$^-$). To test this, we used partial least squares discriminant analysis (PLS-DA) to predict cue type from recall test behavior (no

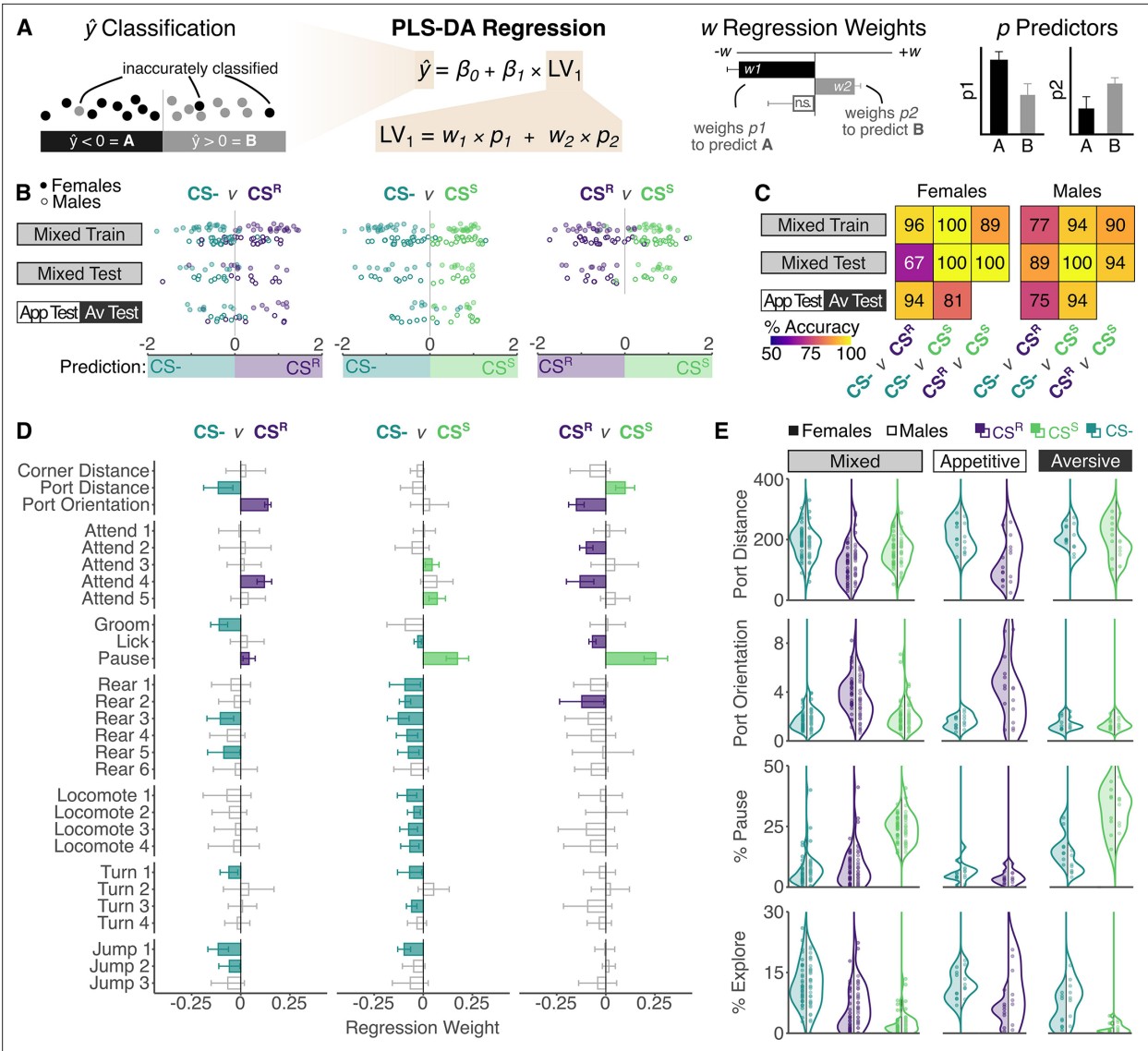

**Figure 2.** Cue identity can be reliably predicted from data-driven analysis approaches. (**A**) Overview of partial least squares discriminant analysis (PLS-DA) approach. (**B**) Predictions of cue type on recall day using two-way classifiers in individual female (filled points) and male (empty points) mice in the mixed training dataset, mixed test dataset, and appetitive- and aversive-only datasets. Point colors indicate true cue type, with predicted cue type indicated by position on x-axis. (**C**) Percent accuracy of cue type prediction in females (left) and males (right). All accuracy values surpass chance prediction (50%). (**D**) Regression weights for individual predictors for each two-way classifier. Significant predictors (p < 0.05) are filled with the color of the cue type an increased value is predictive of, non-significant predictors are gray. Full statistical results are reported in *Supplementary file 1d*. (**E**) Distributions of variables predicting cue types. Port distance is decreased and port orientation is increased during the CS$^R$, pausing is increased during the CS$^S$, and exploration (sum of all rearing and locomoting syllables) is highest during the CS$^-$. Errorbars indicate 95% confidence interval for jackknife estimate. Sample sizes: Mixed: $n_{Male}$ = 37, $n_{Female}$ = 37; Appetitive: $n_{Male}$ = 8, $n_{Female}$ = 8; Aversive: $n_{Male}$ = 8, $n_{Female}$ = 8.

The online version of this article includes the following figure supplement(s) for figure 2:

**Figure supplement 1.** Trajectory plots of individual syllables.

**Figure supplement 2.** Prediction accuracy and component (latent variable) selection.

**Figure supplement 3.** % Freezing correlation with % Pausing in single- or mixed-valence test datasets.

**Figure supplement 4.** Raw values on recall day of all variables included in the partial least squares discriminant analysis (PLS-DA) model not shown in *Figure 2*.

outcomes delivered) (*Figure 2A*, *Figure 2—figure supplement 2*). Two-way classifiers differentiating cue pairs in the mixed paradigm (CS⁻ vs CSᴿ, CS⁻ vs CSˢ, and CSᴿ vs CSˢ) achieved high classification accuracy in both male and female mice, indicating that mice had indeed learned both appetitive and aversive associations (*Figure 2B, C*). Applying the classifiers trained on the mixed paradigm to the single-valence paradigms also yielded above-chance classification accuracy, suggesting that mice express valence responses similarly across mixed and single-valence paradigms.

We then asked which behavioral patterns contribute most strongly to cue type prediction by examining the regression weights of the PLS-DA regression latent variable. Food-port-related location-based metrics, '*port-distance*' and '*port-orientation*', strongly contributed to predicting CSᴿ from CS⁻ (*Figure 2C*). Increased '*grooming*', '*rearing*', and '*locomoting*' behaviors contributed to predicting CS⁻ from CSᴿ. '*Pausing*' behavior most strongly contributed to predicting CSˢ from CS⁻ (*Figure 2C*), and again CS⁻ associated with similar patterns of general movement. Pausing is strongly correlated with '*freezing*' (*Figure 2—figure supplement 3*), suggesting that the pausing syllable is largely capturing what is classically considered '*freezing*'. However, we refrain from labeling the syllable '*freezing*' because '*freezing*' is conceptualized by explicitly defined criteria, such as absence of all movement for a minimum duration, which we do not probe in this data-driven approach (*Blanchard and Blanchard, 1969*). Differentiating CSᴿ from CSˢ again relied on food-port-related location-based metrics for CSᴿ and '*pausing*' for CSˢ, suggesting that these are useful behavioral metrics to identify appetitive and aversive responses. Visualizing raw data for these behavioral predictors confirmed expected distributions across cue types (*Figure 2D*, *Figure 2—figure supplement 4*). Overall, this suggests that the lack of evidence for learning observed with standard behavioral metrics on recall day reflects the limitations of the metrics in a mixed-valence setting rather than a lack of learning.

## Apparent sex differences in learning reflect underlying context-driven exploration

Having built robust behavioral classifiers, we applied these to probe how performance changes in male and female mice over training across paradigms (*Figure 3A*). In the mixed paradigm, both males and females rapidly discriminate CSᴿ from CSˢ, indicating they quickly acquire valence-specific associations. In contrast, discriminating either valenced cue from the CS⁻ varied markedly across paradigms and sexes. By mid-training, females discriminate CSᴿ from CS⁻ in both the mixed and appetitive-only paradigm. In contrast, although males learn to discriminate the CSᴿ from CS⁻ in both paradigms, this occurs later than in females, appearing robustly only toward training end. Uniquely, in males in the appetitive-only paradigm, the CSᴿ is initially misclassified as CS⁻ suggesting an altered trajectory of CSᴿ versus CS⁻ discrimination. Notably, despite these differences, males eventually acquire the CSᴿ association and CS⁻ discrimination in both protocols (*Figure 3B*, *Figure 3—figure supplement 1*).

Intriguingly, predicting CSˢ from CS⁻ revealed distinct effects of sex. Females rapidly discriminate CSˢ from CS⁻ in the mixed paradigm, but in the aversive-only paradigm, the CS⁻ is consistently misclassified as CSˢ throughout training, with discrimination only evident at recall (*Figure 3A, B*). In contrast, males similarly discriminate CSˢ from CS⁻ in both paradigms. The misclassification of CSᴿ as CS⁻ in males and CS⁻ as CSˢ in females suggests that males are exhibiting more CS⁻-associated behaviors ('*rearing*' and '*locomoting*') during the CSᴿ and that females are showing less of these same behaviors during the CS⁻ and more of the CSˢ-associated *pausing* behaviors. Integrating these observations led us to hypothesize that a common underlying factor of exploration might explain the observed sex differences in cue responding in single-valence protocols.

We reasoned that, if exploration drives sex-specific cue responding, sex differences in exploration should be apparent even before training begins. Specifically, we predicted that males explore more than females. To test this, we compared exploration on the habituation day and pre-cue periods during training by quantifying the total percent time in any *rearing* or *locomoting* syllable (from here on summarized as '*exploring*'; *Figure 3C*). This confirmed that males explore more than females throughout the habituation day when no valenced outcomes have been presented. During the pre-cue period, there were no sex differences in exploration in the mixed-valence paradigm. However, as predicted, in the appetitive-only paradigm, males explored more than females, and in the aversive-only paradigm, this trend was maintained. To confirm that elevated exploration inhibits the expression of valence-specific behaviors, we correlated these metrics, confirming the predicted negative relationships (*Figure 3D*). Overall, this demonstrates that behavioral expression in single-valence paradigms

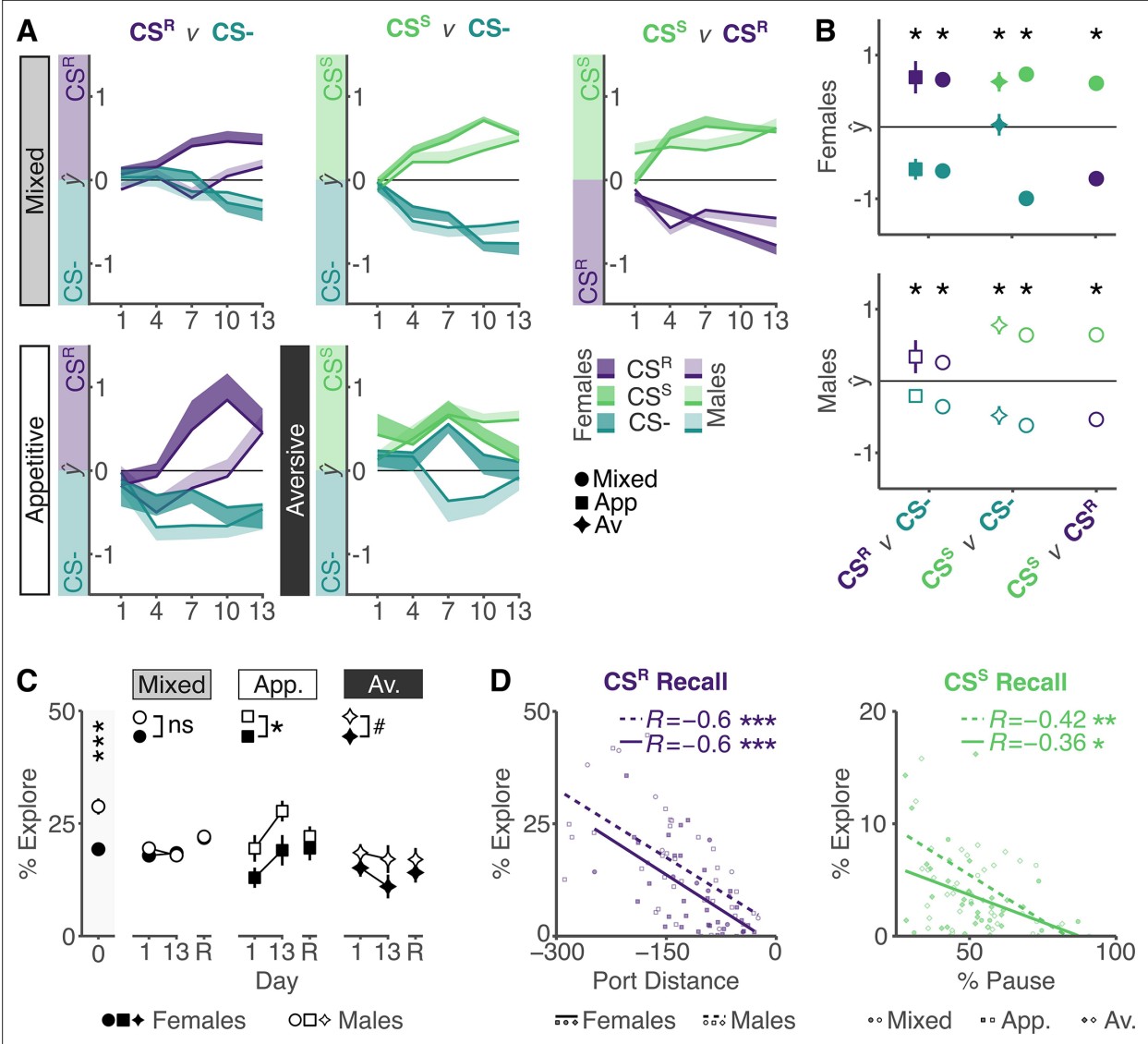

**Figure 3.** Performance trajectories are mediated by underlying sex differences in exploration. (**A**) Cue type prediction throughout training in the mixed paradigm (top) and single-valence paradigms (bottom) throughout training in each two-way classifier. Male and female predictions largely follow each other in the mixed paradigm, whereas sex-specific learning trajectories can be observed in both the appetitive and aversive-only paradigm. (**B**) Cue type prediction on recall day for each paradigm in females (top) and males (bottom) shows cue discrimination between all cue types in both sexes (<u>contrast CS--CSR</u>: appetitive female: $t\text{-ratio}_{360} = -4.635$, $p_{adj} < 0.001$; mixed female: $t\text{-ratio}_{360} = -7.92$, $p_{adj} < 0.001$; appetitive male: $t\text{-ratio}_{360} = -0.547$, $p_{adj} < 0.05$; mixed male: $t\text{-ratio}_{360} = -3.629$, $p_{adj} < 0.001$; <u>contrast CS--CSS</u>: aversive female: $t\text{-ratio}_{360} = -2.165$, $p_{adj} < 0.05$; mixed female: $t\text{-ratio}_{360} = -15.1$, $p_{adj} < 0.001$; aversive male: $t\text{-ratio}_{360} = -4.544$, $p_{adj} < 0.001$; mixed male: $t\text{-ratio}_{360} = -11.746$, $p_{adj} < 0.001$; <u>contrast CSR--CSS</u>: mixed female: $t\text{-ratio}_{360} = -7.18$, $p_{adj} < 0.001$; mixed male: $t\text{-ratio}_{360} = -8.117$, $p_{adj} < 0.001$). The full statistical model is reported in ***Supplementary file 1e***. (**C**) Percent time spent exploring in the pre-cue period during habituation, first day of training (day 1), end of training (day 13), and on recall test (R). Males and females differ in their levels of exploration on habituation (<u>contrast female–male</u>: $t\text{-ratio}_{30} = -4.281$, $p < 0.001$), and throughout training in the single-valence paradigms (<u>contrast female–male</u>: appetitive: $t\text{-ratio}_{85.9} = -2.559$, $p < 0.05$; aversive: $t\text{-ratio}_{88.1} = -1.784$, $p = 0.078$) but not the mixed-valence paradigm (<u>contrast female–male</u>: $t\text{-ratio}_{94.3} = -0.373$, $p = 0.71$). The full statistical model is reported in ***Supplementary file 1f***. (**D**) Cue-relevant behaviors port distance (CSR predictive, left) and pausing (CSS predictive, right) negatively correlate with % exploration during cue presentation. *$p_{adj} < 0.05$, **$p_{adj} < 0.01$, ***$p_{adj} < 0.001$. Errorbars and ribbons indicate SE. Sample sizes: Mixed: $n_{Male} = 37$, $n_{Female} = 37$; Appetitive: $n_{Male} = 8$, $n_{Female} = 8$; Aversive: $n_{Male} = 8$, $n_{Female} = 8$.

The online version of this article includes the following figure supplement(s) for figure 3:

**Figure supplement 1.** CSR predictive behaviors in females (top) and males (bottom).

is biased by sex differences in exploration. This bias is not present in a mixed-valenced paradigm, suggesting that a mixed-valenced paradigm provides a more accurate assessment of valence learning. Performance in the mixed-valenced paradigm demonstrates that both sexes *can* similarly learn valenced associations, yet it is not clear why females fail to do this in the aversive-only context. To understand why sex differences in exploration should be specific to this single-valence paradigm, we interrogated the specific conditions under which these sex differences are expressed.

### Shock exposure broadly facilitates *pausing* to suppress exploration in female mice

To gain insight into paradigm-specific sex differences in exploration, we compared exploration across the mixed-valence and aversive-only paradigms. We calculated the percent time spent in exploration or the relative absence of exploration, defined as '*pausing*', in the pre-cue period to determine if context modulates behavior independently of cues. In females, but not in males, levels of exploration and pausing differed between paradigms with overall higher exploration and lower pausing in the mixed paradigm (*Figure 4A*). This suggests that the outcomes experienced during training and the associative context modulate baseline exploration behavior in females more strongly than in males.

In the aversive-only paradigm, we demonstrated that high levels of '*pausing*' to CS⁻ obscure CSˢ discrimination (*Figure 3A*). Having now shown that high levels of pausing are also observed in the pre-cue period, we asked if the elevated freezing to CS⁻ is cue-driven or reflects a continuation of elevated context-mediated freezing. To test this, we correlated *exploration* and *pausing* between the pre-cue and cue periods for CS⁻ and CSˢ. We reasoned that, if cue onset elicits a distinct behavioral response, behavior between pre-cue and cue periods will not correlate. Indeed, for the CSˢ, we observed no significant pre-cue to cue correlation in either paradigm or sex (*Figure 4B*). However, for the CS⁻, we observed consistent, strong correlation across paradigms and sexes indicating that behavior during the CS⁻ cue period is a continuation of contextually mediated behavior and not a distinct cue-mediated response.

Altered pre-cue responding likely reflects the combination of the acute impact of innately appetitive or aversive outcomes encountered during the current session and the evolving associative history of the context. Both factors contribute on training days, whereas on the recall day, only the associative history of the context is present. In the aversive-only paradigm, while females failed to show evidence of CS⁻ versus CSˢ discrimination during training, this was clearly evidenced on the recall day (*Figure 3B*). We therefore hypothesized that pre-cue differences are in part attributable to acute effects of outcome presentations. We reasoned that this effect will be amplified across the session as the impact of successive outcomes accrues but would not be apparent on the non-reinforced recall session. To test this, we compared *pausing* during the pre-cue, CSˢ, and CS⁻ periods for early (first two) and late (final two) cue presentations on training days 1 and 13 and recall (*Figure 4C–E*). In the mixed paradigm, as expected, male and female mice *paused* more in the late pre-cue periods on both training day 1 and 13 but not during recall, suggesting a modest influence of outcomes. In the aversive-only paradigm, this modulation was stronger in both sexes on training day 1; however, by training end, this was no longer evident in males, but females continued to show elevated *pausing* across the session (*Figure 4C*). This suggests that males, but not females, habituated to footshock, downregulating outcome-induced pausing across training days. As predicted, in females, this phenomenon was also evident in CS⁻ responding (*Figure 4D*), but no such difference was observed for the CSˢ (*Figure 4E*). Overall, this suggests that, in females, shock exposure acutely potentiates *pausing* specifically in the aversive-only paradigm and that this obscures CS⁻ discrimination. This demonstrates that task-dependent sex differences in non-associative learning processes can mask similar underlying associative learning.

## Discussion

Despite the fundamental importance of valence learning to health and disease, limited research has simultaneously explored associative valence learning in male and female mice. Here, we compared mixed- and single-valence paradigms and revealed critical effects of task design on exploration and habituation, a form of non-associative learning, that interact with sex to alter behavioral expressions of learning. We demonstrate that both male and female mice can successfully acquire appetitive

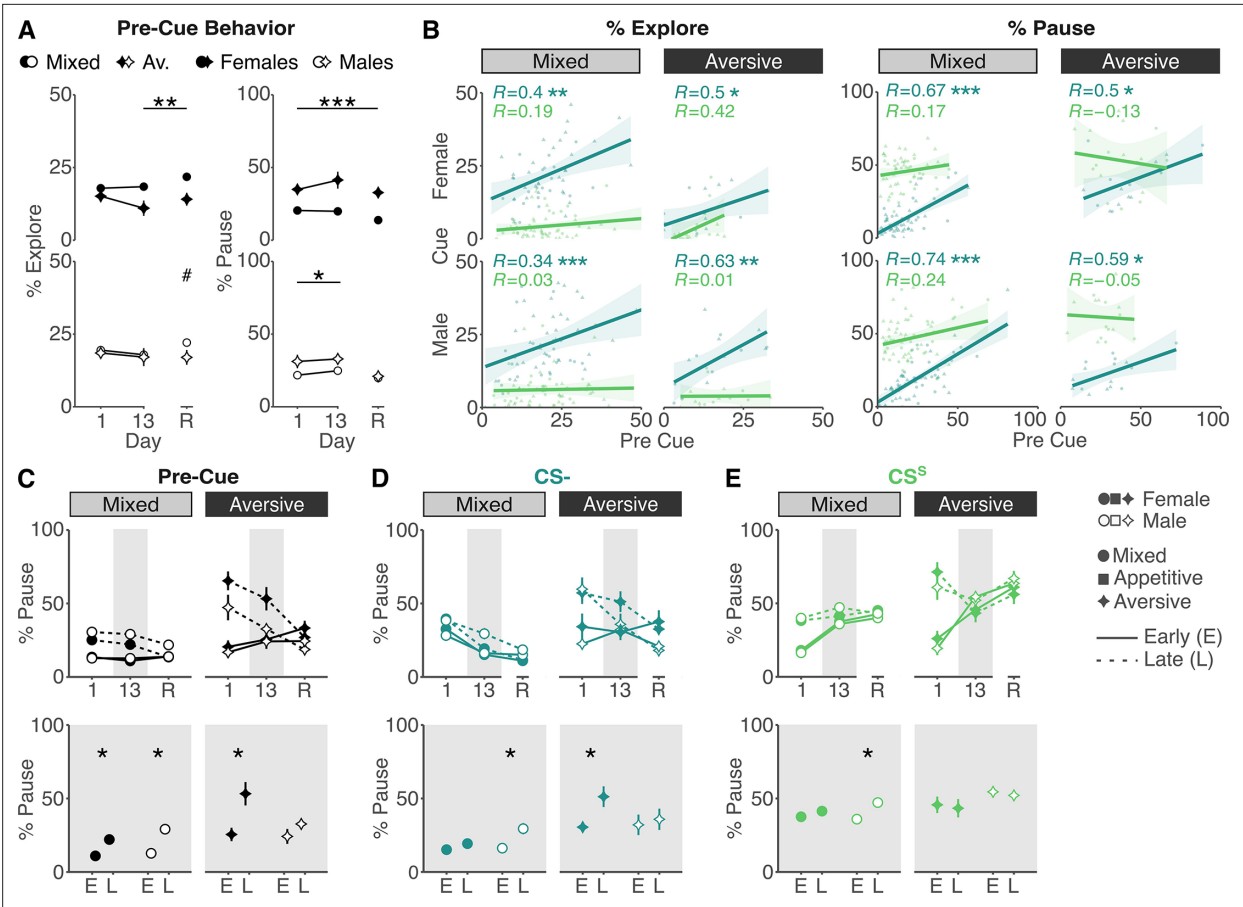

**Figure 4.** Repeated shock exposure promotes widespread pausing, limiting exploratory behavior in female mice. (**A**) Pre-cue exploration and pausing behaviors show significant differences between mixed-valence and aversive-only paradigms throughout training and recall in females (top; contrasts explore mixed vs aversive: all days: $t\text{-ratio}_{77.5} = -6.201$, $p_{adj} < 0.001$; day 1: $t\text{-ratio}_{195} = -3.548$, $p_{adj} < 0.001$; day 13: $t\text{-ratio}_{191} = -5.219$, $p_{adj} < 0.001$; day R: $t\text{-ratio}_{186} = -4.81$, $p_{adj} < 0.001$; contrasts pause mixed vs aversive: all days: $t\text{-ratio}_{77} = 3.338$, $p_{adj} < 0.01$; day 1: $t\text{-ratio}_{213} = 1.025$, $p_{adj} = 0.306$; day 13: $t\text{-ratio}_{212} = 2.7$, $p_{adj} < 0.01$; day R: $t\text{-ratio}_{210} = 2.886$, $p_{adj} < 0.01$). In males (bottom), time spent exploring is comparable across paradigms, and time spent pausing is only modestly increased in the aversive-only paradigm during training (contrasts exploring mixed vs aversive: all days: $t\text{-ratio}_{73.6} = -2.159$, $p_{adj} < 0.05$; day 1: $t\text{-ratio}_{185} = -2.419$, $p_{adj} < 0.05$; day 13: $t\text{-ratio}_{187} = -2.051$, $p_{adj} < 0.05$; day R: $t\text{-ratio}_{185} = -0.288$, $p_{adj} = 0.774$; contrasts pausing mixed vs aversive: all days: $t\text{-ratio}_{72} = 1.288$, $p_{adj} = 0.202$; day 1: $t\text{-ratio}_{210} = 0.367$, $p_{adj} = 0.714$; day 13: $t\text{-ratio}_{210} = 0.277$, $p_{adj} = 0.782$; day R: $t\text{-ratio}_{210} = 1.911$, $p_{adj} = 0.057$). The full statistical model is reported in **Supplementary file 1g**. (**B**) Correlations between $CS^-$ and $CS^S$ pre-cue and cue exploration (left) and pausing (right) behaviors show that $CS^-$ behaviors correlate significantly with pre-cue behaviors, whereas $CS^S$ behaviors do not. (**C**) Percent time spent pausing increases over the course of the session from the first two cues (early; E) to the last two cues (late; L) in the pre-cue period on the first day of training in both paradigms and sexes (contrasts day 1 early vs late: females mixed: $t\text{-ratio}_{360.7} = -3.381$, $p_{adj} < 0.001$; females aversive: $t\text{-ratio}_{360.7} = -6.097$, $p_{adj} < 0.001$; males mixed: $t\text{-ratio}_{360.7} = -5.249$, $p_{adj} < 0.001$; males aversive: $t\text{-ratio}_{360.7} = -4.424$, $p_{adj} < 0.001$). On day 13 (resolved in bottom plot), both male and female mice show small but significant increases in pausing at the end of the session while, in the aversive-only paradigm, females show a large increase in pausing at the end of the session absent in males (contrasts day 13 early vs late: females mixed: $t\text{-ratio}_{360.7} = -2.802$, $p_{adj} < 0.01$; females aversive: $t\text{-ratio}_{360.7} = -4.023$, $p_{adj} < 0.001$; males mixed: $t\text{-ratio}_{360.7} = -4.613$, $p_{adj} < 0.001$; males aversive: $t\text{-ratio}_{360.7} = -1.229$, $p_{adj} = 0.22$). On recall day (R), pausing levels are only significantly elevated at the end of the session in males in the mixed paradigm (contrasts day R early vs late: females mixed: $t\text{-ratio}_{360.7} = 0.113$, $p_{adj} = 0.91$; females aversive: $t\text{-ratio}_{360.7} = 0.921$, $p_{adj} = 0.353$; males mixed: $t\text{-ratio}_{360.7} = -2.355$, $p_{adj} < 0.05$; males aversive: $t\text{-ratio}_{360.7} = 0.832$, $p_{adj} = 0.406$). (**D**) Similarly, percent time spent pausing to the $CS^-$ increases throughout the session on day 13 in females in the aversive-only paradigm, with a more modest increase in males in the mixed-valence paradigm (contrasts day 13 early vs late: females mixed: $t\text{-ratio}_{356.5} = -0.925$, $p_{adj} = 0.355$; females aversive: $t\text{-ratio}_{356.5} = -2.694$, $p_{adj} < 0.01$; males mixed: $t\text{-ratio}_{360.7} = -3.332$, $p_{adj} < 0.001$; males aversive: $t\text{-ratio}_{360.7} = -0.492$, $p_{adj} = 0.623$). (**E**) On day 13 levels of pausing are stable from early to late session, except for a modest increase in males in the mixed paradigm (contrasts day 13 early vs late: females mixed: $t\text{-ratio}_{356.7} = -0.96$, $p_{adj} = 0.338$; females aversive: $t\text{-ratio}_{356.7} = 0.335$, $p_{adj} = 0.738$; males mixed: $t\text{-ratio}_{357.9} = -3.164$, $p_{adj} < 0.01$; males aversive: $t\text{-ratio}_{356.7} = 0.347$, $p_{adj} = 0.729$). Significance levels are shown for day 13 comparisons only (bottom plots), with full statistical results reported in **Supplementary file 1h**. $^{\#}p_{adj} < 0.1$, $^{*}p_{adj} < 0.05$, $^{**}p_{adj} < 0.01$, $^{***}p_{adj} < 0.001$. Errorbars indicate SE. Sample sizes: Mixed: $n_{Male} = 37$, $n_{Female} = 37$; Appetitive: $n_{Male} = 8$, $n_{Female} = 8$; Aversive: $n_{Male} = 8$, $n_{Female} = 8$.

and aversive cue–outcome associations across mixed- and single-valence tasks. Using data-driven approaches to interrogate cue-mediated behavioral changes, we identify robust performance metrics of cue–outcome associations that are applicable across paradigms and sexes. Critically, we show that behavior in single-valence paradigms is biased by pre-existing sex differences in exploratory behaviors that mask behavioral expression of cue learning, potentially leading to erroneous conclusions of sex differences in associative valence learning. Our analyses demonstrate that performance in the mixed-valence protocol is less biased by baseline sex differences in exploratory behaviors, thereby allowing more accurate assessment of learning.

Our findings shed new light on previously reported sex differences in single-valence learning tasks. Fear conditioning studies report stronger threat generalization and higher levels of conditioned responding in female rodents (*Day et al., 2016*; *Day et al., 2020*; *Olivera-Pasilio and Dabrowska, 2023*; *Trott et al., 2022b*). Superficially, we replicate these findings in an aversive-only protocol, but comparison to the mixed-valence protocol suggests that these observations do not reflect an inherent inability of female mice to discriminate between the CS$^S$ and the CS$^-$. Rather, we show that the lack of discrimination is driven by lack of habituation to repeated shock, a form of non-associative learning (*Ardiel and Rankin, 2010*), and pre-existing sex differences in the propensity to engage in exploration. Research in rats has previously shown darting as a female-specific sexually divergent response in fear conditioning, a feature we did not observe in our data, possibly due to species or apparatus differences (*Gruene et al., 2015*; *Mitchell et al., 2024*; *Mitchell et al., 2022*) or CS characteristics (*Borkar et al., 2024*; *Borkar et al., 2020*; *Dong et al., 2019*; *Fadok et al., 2017*; *Le et al., 2024*; *Trott et al., 2022a*). Research in reward learning, while sparser, has also suggested sex differences, with male rodents showing less robust reward conditioning (*Lefner et al., 2022*). Here, we demonstrate that apparent sex differences in single-valence learning tasks are driven by underlying differences in exploration that mask cue learning in each sex depending on context and are not true learning differences. The underlying neurobiological causes for these sex-specific effects of context on behavior will be an interesting target for future research.

Sex differences in exploratory behavior in mice have previously been reported in other tasks. For example, in a reward-based decision-making task, males had overall higher levels of exploration than females, but females were able to learn from exploration more quickly (*Chen et al., 2021b*). Curiously, other research suggests increased velocity or rearing in female rodents compared to males in the open field test or social interaction test (*McElroy and Howland, 2025*; *Schuler et al., 2025*). This indicates that sex differences in exploration may be task specific with increased exploration in females in anxiety-inducing contexts, consistent with reports that increased locomotion is a female-specific anxiety-like behavior (*Gruene et al., 2015*; *Mitchell et al., 2022*; *Schuler et al., 2025*). Overall, this study adds to the growing literature on the nuances of sex differences in behavior (*Chen et al., 2021a*; *Greiner et al., 2019*; *Gruene et al., 2015*; *Pellman et al., 2017*; *Shansky, 2018*; *Tronson, 2018*).

Our analysis highlights the importance of task design in shaping behavioral expression. We find elevated periods of immobility in the aversive-only paradigm compared to the mixed-valence paradigm for both sexes, suggesting an increased influence of contextual learning in single-valence fear learning. This is consistent with the varying predictive value of contexts between single and mixed-valence paradigms. In the single-valence paradigm, the only salient outcome that is encountered is foot shock, making the context a relatively good predictor of outcome. In contrast, in the mixed-valence paradigm, two salient, oppositely valenced outcomes are encountered – footshock and reward – rendering the context a poor predictor of outcome. In this way, mixed-valence contexts reduce contextual learning and better isolate cue-modulation of behavior, limiting the influence of pre-existing sex differences in exploration on behavioral expression. Reduced habituation to repeated shock in female mice was also task dependent and was restricted to the aversive-only setting. This suggests a female-specific sensitivity to repetitive aversive experiences, suggesting a potentially important sex-specific phenotype for future studies with relevance to human sex differences in neuropsychiatric conditions. Our findings emphasize the need for caution in interpreting sex differences in task performance and illustrate the impact of task design.

In line with previous research, our findings also emphasize the power of rigorously applied data-driven approaches in behavioral neuroscience in re-evaluating long-standing assumptions and identifying sex-specific phenotypes in behaviors previously established exclusively on male data (*Levy et al., 2023*; *Schuler et al., 2025*; *Shansky, 2024*). Our data show that relying exclusively on existing

behavioral metrics can be misleading when developing a novel behavioral protocol. Data-driven approaches, as the one employed in this study, remove biased and preconceived notions from the interpretation of animal behavior, leading to identification of behaviors solely revealed from the data structure with minimal human interference. PLS-DA provides a novel way to understand and interpret the high-dimensional data produced by pose estimation and behavioral classifiers. Behavioral classifiers can identify a large number of syllables, and rigorously comparing the duration spent in each syllable between outcomes of interest is not feasible due to the large amount of multiple testing corrections required in combination with usually small sample sizes in behavioral neuroscience experiments. As compared to unsupervised dimensionality reduction methods, such as principal component analysis, PLS-DA identifies axes of variance in the predictors that are correlated with the outcome. This makes it a useful approach in that it explicitly captures features that play an important role in outcome prediction. We present a blueprint for rigorously identifying relevant behavioral metrics of learning in both sexes that can be applied across a wide variety of paradigms, including animal models of neuropsychiatric disease.

We established a mixed-valence paradigm with robust behavioral readouts in male and female mice. While similar tasks have been previously designed in rats or in mouse head-fixed setups, it has been challenging to study approach and avoidance behavior in a within-subject design in freely moving mice. We note that, specifically for approach behavior, standard readouts such as head entries or licks perform poorly in quantifying individual learning. This challenge is perhaps reflected in the scarcity of publications that report cue-based reward learning in freely moving mice. We observed that in both appetitive-only and mixed-valence paradigms, mice do not physically enter the food port until reward delivery, yet other approach behaviors are evident in video-based quantification. Using advanced behavioral classification to overcome this challenge, we identified simple behavioral measures that can be easily applied in future using simpler tracking methods (e.g., ezTrack). The mixed-valence paradigm effectively isolates valence learning and overcomes limitations of single-valence approaches to model real-world learning in which mixed-valence outcomes are commonly encountered. However, several important task parameters that influence learning and behavior remain to be further explored, including varying the US modality and US salience (*Mitchell et al., 2024*). Further, the precise setup of the conditioning apparatus will likely also influence task performance. For example, our use of a non-retractable food port allowed mice to consume chocolate milk even after the cue ends, potentially slowing learning compared to a retractable reward delivery apparatus, in which mice can only consume chocolate milk during the cue period. Future work could fully explore the parameter space in appetitive- and aversive-only protocols to fully understand the boundaries of sex-specific effects.

To conclude, we demonstrate that both males and females are able to acquire appetitive and aversive associations in both single- and mixed-valence paradigms, yet the expression of this learning is shaped by task design. Our findings emphasize the critical importance of robust behavioral phenotyping in both sexes to achieve accurate insight into learning. We compared single- and mixed-valence paradigms and revealed how task variables and experimental design considerations differentially impact behavioral expression in male and female mice in cued valence learning. Most importantly, we show that learning assessed in single-valence paradigms is biased by sex differences in exploration and that mixed-valence contexts minimize these non-task relevant sex differences to provide an unobscured assessment of learning in both sexes. Leveraging mixed-valence learning protocols in combination with in vivo circuit interrogation techniques and animal models for disease promises to provide clear insight into the mechanisms of valence processing in health and disease.

## Materials and methods

### Animals

Seven-week-old male and female C57BL/6J mice (Jackson laboratories, Bar Harbor, ME, USA) were group-housed by sex ($n$ = 5/cage) for 1 week prior to experiments. All mice were maintained at 22–25°C, on a 12-hr light/dark cycle (lights on at 07:00 hr), with ad libitum food and water prior to the beginning of the experiments. Animals were food restricted at the start of conditioning and maintained at 85% body weight. All testing occurred throughout the light cycle. All procedures were approved by the Animal Care Committee and conformed to McGill University Comparative Medicine and Animal Resources Centre guidelines.

## Apparatus

Mice were trained in conditioning boxes (15.24 × 13.34 × 12.7 cm; Med Associates Inc, USA) enclosed in sound-attenuating chambers outfitted with a programmable audio generator connected to a speaker, a grid floor, an infrared light, a fan for white noise and ventilation, and a food port for liquid reward delivery through a syringe pump. Each food port is equipped with an infrared beam to quantify head entries and a lickometer to quantify licks. Boxes were controlled and data collected by a computer running MED-PC software (Med-Associates Inc, USA).

Sessions were recorded from top view using AniHome (*Singh et al., 2019*) on Raspberry Pi (Raspbian GNU/Linux 10 (buster)) with infrared camera modules (Raspberry Pi Camera Module 3 NoIR 3) at 800 × 600 pixels resolution and 40 frames per second. To allow for video-based identification of cue periods, infrared LEDs mounted on top of the operant box were triggered each time a cue was presented. Metal trays underneath the grid floor were filled with 1/8″ corncob bedding to prevent reflections of infrared lighting in the recording.

## Classical conditioning protocols

### Task design considerations

Mice take longer to learn an appetitive association than an aversive association (*Deseyve et al., 2024*). Repeated exposure to cue–outcome pairings after the association has been acquired can lead to overtraining and habituation in animals, such that a learned behavioral response (e.g., freezing) may no longer be exhibited to the cue (*Carroll et al., 2024*; *Tafreshiha et al., 2021*). To balance these considerations, protocols incorporated more appetitive cue–outcome pairings than aversive cue–outcome pairings. Single-valence conditioning protocols are designed as a subset of the mixed-valence conditioning protocol to support direct comparison between the paradigms and account for differences in training duration between appetitive and aversive single-valence conditioning protocols.

### Conditioning protocols

Animals were transported from the colony to the testing room in their home cage and left to habituate under red light 1 hr prior to session start. Conditioning was preceded by a single day of context and cue habituation, during which mice were placed in the operant box and exposed to five presentations of each cue without outcomes. Each conditioning session began with a 2-min habituation period, followed by presentations of auditory cues (CS) in random order separated by a variable inter-trial interval (average 180 s). The appetitive CS (CS Reward; $CS^R$) was paired with 30 µl of chocolate milk dispensed into the food port. The aversive CS (CS Shock; $CS^S$) was paired with a 0.5-mA foot shock delivered through the grid floor for 0.5 s. The $CS^-$ was not paired with any outcome. Auditory cues were presented for 15 s, with outcomes delivered randomly within the final 5 s of the cue. Cues were reinforced 80% of the time, as prediction error improves learning (*Iordanova et al., 2021*). Boxes were cleaned and bedding replaced between animals. One day after completion of conditioning, mice were exposed to a recall session (five presentations of each cue without outcomes) to assess behavioral responses to cues in the absence of outcomes.

### Mixed-valence conditioning

Training began with a mixed-valence training session, followed by two appetitive-only training sessions, repeating this order for 14 daily training sessions. Mixed-valence training sessions consisted of 10 $CS^R$, 5 $CS^S$, and 5 $CS^-$ presentations. Appetitive-only training sessions consisted of 10 $CS^R$ and 10 $CS^-$. Cue identity was counterbalanced for $CS^R$ and $CS^S$ between two pure tones (2 kHz, 75 dB or 10 kHz, 63 dB), while $CS^-$ was always a clicker (10 Hz, 75 dB).

### Appetitive single-valence conditioning

Appetitive conditioning mirrored appetitive-only training days in the mixed-valence conditioning protocol. Training lasted for 14 daily sessions. Each session consisted of 10 $CS^R$ and 10 $CS^-$. Cue identity was counterbalanced between mice for $CS^R$ between two pure tones (2 kHz, 75 dB or 10 kHz, 63 dB), while $CS^-$ is a clicker (10 Hz, 75 dB).

## Aversive single-valence conditioning

Aversive conditioning mirrored the aversive components of mixed training days in the mixed-valence conditioning protocol. Training lasted for 14 days with training sessions every third day interspersed with two rest days when mice remained in their home cage. Each session consisted of 5 $CS^S$ and 5 $CS^-$. Cue identity was counterbalanced between mice for $CS^S$ between two pure tones (2 kHz, 75 dB or 10 kHz, 63 dB), while $CS^-$ is a clicker (10 Hz, 75 dB).

## Conditioning cohorts

The mixed-valence discovery and validation cohort consisted of 56 ($n_{Male}$ = 28, $n_{Female}$ = 28) and 18 ($n_{Male}$ = 9, $n_{Female}$ = 9) mice, respectively. The single-valence cohorts consisted of 16 animals each ($n_{Male}$ = 8, $n_{Female}$ = 8).

## Behavioral analysis

### Data analysis

Behavior was evaluated using video-based approaches as well as food-port generated outputs. Whole session video recordings were cropped into 45 s clips centered on the cue interval (15 s prior to cue onset to 15 s post cue offset) for freezing analysis (ezTrack) and detailed behavioral profiling (Deep-LabCut and Keypoint-MoSeq). Head entries, head exits (infrared beam), and licks (contact lickometer) were recorded and analyzed as indicators of reward seeking. Behavior was quantified exclusively in the first 10 s of the cue (cue window; i.e., prior to outcome delivery).

### ezTrack

Freezing was quantified using ezTrack (*Pennington et al., 2021*; *Pennington et al., 2019*) in the first 10 s of the cue (cue window; i.e., prior to outcome delivery). To confirm that our findings are robust across thresholding decisions, we explored the following parameter spaces: minimum duration = 0.25, 0.5, and 1 s; motion threshold = 10, 25, 50, and 100 px.

### DeepLabCut

DeepLabCut (v2.2.1.1; *Lauer et al., 2022*; *Mathis et al., 2018*; *Nath et al., 2019*) was used for markerless pose estimation. 95% of all labeled frames were used to train a ResNet-50 based neural network with default parameters for 500,000 training iterations. The train error was 4.48 pixels, and the test error was 5.84 pixels. The resulting neural network was used to track body parts for all cue-interval clips.

### Keypoint-MoSeq

Keypoint-MoSeq (kp-MoSeq; v0.5.0; *Weinreb et al., 2024*) was used for unsupervised behavioral classification. Sixteen cue clips for each cue type were semi-randomly selected from each session over 14 training and 1 recall session of the training mixed-valence dataset. The model was fit over 50 iterations in the autoregressive only model and over an additional 450 iterations in the full model. Kappa was tuned to 1e5 for the autoregressive-only model and 5e4 for the full model to achieve a median syllable duration of 400 ms as recommended by *Weinreb et al., 2024*.

kp-MoSeq uses a stochastic fitting procedure that will produce slightly different syllable segmentations when run multiple times with different random seeds. For this reason, 20 kp-MoSeq models were generated using 20 different seeds and the best model was selected based on maximal expected marginal likelihood.

### Syllable post-processing

Twenty-four syllables were generated and manually labeled by an experimenter (*Supplementary file 1c*; *Figure 2—figure supplement 1*) into one of the following categories to comprehensively profile behavior and support interpretability: attend, jump, rear, locomote, turn, and other (groom, pause, and lick). Syllable cumulative duration (referred to as 'duration') was quantified as the sum of frames that a syllable was present during a given time window.

## Location-based predictors

We included location-based predictors to incorporate spatially salient points in the prediction model. We reasoned that the food port and corners of the conditioning chambers define salient locations during reward seeking and threat anticipation, respectively. Distance to the food port was calculated as distance from snout to food port. Food port orientation was calculated as follows: distance from tail base to food port/distance from snout to food port. Thus, values above 1 indicate closer snout distance (i.e., port orientation), whereas values below 1 indicate closer tail base distance (i.e., orientation away from port). The distance of the tail base to each corner was calculated, and the smallest distance included as a predictor. All distances were measured in pixels.

## Statistical analysis

### Linear mixed-effects regression

Repeated measures data were modeled using linear mixed-effects regression. Mouse ID was added as a random effect in all models to account for interdependence of observations within individuals across days. Planned contrasts were performed to determine statistical significance of comparisons using each model's estimated marginal means. Multiple comparisons were performed using Tukey's HSD to control the family-wise error rate.

### Partial least squares discriminant analysis

PLS-DA is a supervised statistical approach used for classification and dimensionality reduction (*Rosipal and Krämer, 2006*). Using syllable frequencies as predictors makes them inherently correlated, because animals can only be one syllable at a time. In addition, the location-based predictors are likely correlated with syllable frequency, as some syllables may be performed preferentially at specific locations within the condition box. PLS-DA overcomes this challenge in the data structure, as it performs well with large numbers of correlated predictors and inherently provides classification and feature selection.

Here, we used PLS-DA to predict CS identity from syllable duration and location-based predictors on the recall day data. We used data from the discovery cohort as training data and data from a separate validation cohort as test data to assess prediction accuracy. Model performance was evaluated for two- and three-way classification problems (*Figure 2—figure supplement 2*). Two-way classification consistently outperformed three-way classification, and we therefore used two-way classification models for all downstream analyses. We further compared the performance of prediction using different cue windows and found that 5 s mid-cue performed best. In addition, we compared sex-specific classifiers to each other as well as to classifiers using data of both sexes combined (*Figure 2—figure supplement 2*).

### Software

All analyses were performed in R (v4.3.1). Linear mixed-effects regression was fit using the *lme4* package (v1.1.35.5). Planned contrasts were performed using the *emmeans* package (v1.8.9). PLS-DA was performed using the *mdatools* package (v0.14.2). All data is available at https://osf.io/4xfz2/. All analysis code is available at https://github.com/heike-s/ValenceProfile (copy archived at *Schuler, 2025*).

## Acknowledgements

HS was supported by FRQS. Research was supported by CIHR Project Grants (#180648 and #195897), and a Canada Research Chair to RCB.

## Additional information

### Funding

| Funder | Grant reference number | Author |
|---|---|---|
| Canadian Institutes of Health Research | #180648 | Rosemary C Bagot |
| Fonds de Recherche du Québec - Santé | | Heike Schuler |
| Canadian Institutes of Health Research | #195897 | Rosemary C Bagot |

The funders had no role in study design, data collection, and interpretation, or the decision to submit the work for publication.

### Author contributions

Heike Schuler, Conceptualization, Data curation, Formal analysis, Investigation, Visualization, Methodology, Writing – original draft, Project administration, Writing - review and editing; Eshaan S Iyer, Gabrielle Siemonsmeier, Conceptualization, Methodology; Ariel Mandel Weinbaum, Peter Vitaro, Shiqing Shen, Investigation; Rosemary C Bagot, Conceptualization, Supervision, Funding acquisition, Writing – original draft

### Author ORCIDs

Heike Schuler ● https://orcid.org/0000-0002-8462-5491
Gabrielle Siemonsmeier ● https://orcid.org/0009-0000-9194-7862
Ariel Mandel Weinbaum ● https://orcid.org/0009-0008-9009-2979
Rosemary C Bagot ● https://orcid.org/0009-0005-5512-5294

### Ethics

All procedures were approved by the Animal Care Committee and conformed to McGill University Comparative Medicine and Animal Resources Centre guidelines.

### Decision letter and Author response

Decision letter https://doi.org/10.7554/eLife.108498.sa1
Author response https://doi.org/10.7554/eLife.108498.sa2

## Additional files

### Supplementary files

Supplementary file 1. Supplementary tables for statistics and syllable description. (A) Linear mixed-effects regression (LMER) on food-port-related metrics. (B) LMER on freezing with different thresholds. (C) Syllable identities and description. (D) Jackknifing statistics for regression coefficients for PLS models. (E) LMER comparing predicted values from PLS model on recall day to assess cue discrimination. (F) LMER on Pre-Cue exploring including habituation day. (G) LMER on Pre-Cue pausing and exploring. (H) LMER on pausing in early and late cue by day, sex and paradigm with contrasts for pre-cue, $CS^-$, and $CS^S$. (I) LMER on reward predictive behaviors.

MDAR checklist

### Data availability

All data is available at https://osf.io/4xfz2/. All code for generating figures and statistical analyses is available at https://github.com/heike-s/ValenceProfile (copy archived at *Schuler, 2025*).

The following dataset was generated:

| Author(s) | Year | Dataset title | Dataset URL | Database and Identifier |
|---|---|---|---|---|
| Schuler H | 2025 | Sex-specific exploration accounts for differences in valence learning in male and female mice | https://osf.io/4xfz2/ | Open Science Framework, 4xfz2 |

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
