## [Editor Report]

This important manuscript provides a comprehensive analysis of male and female mice across aversive, appetitive, and mixed-valence Pavlovian conditioning. Using pose estimation, the authors show that standard behavioural measures capture only a fraction of the substantial behavioural reorganization that occurs. They present convincing evidence that animals face competing demands to either respond to cues or explore the environment but cannot do both simultaneously. These findings will be of interest to behavioural neuroscientists and learning theorists studying sex differences.

---

## [Decision Letter]

**Decision letter after peer review:**

Thank you for submitting your article "Sex-specific exploration accounts for differences in valence learning in male and female mice" for consideration by *eLife*. Your article has been reviewed by 3 peer reviewers, and the evaluation has been overseen by a Reviewing Editor and Kate Wassum as the Senior Editor.

*Reviewer #1 (Recommendations for the authors):*

Strengths:

Highly in-depth behavioural assessments, with multiple behaviours measured and compared to assess aversive and appetitive learning.

Interesting data investigating sex differences in baseline exploration – something that is often overlooked.

The fact that findings highlight the need to take into consideration how tasks are designed when interpreting studies that demonstrate sex differences or, really, learning differences of any kind.

Well written overall.

Interesting and insightful discussion.

Weaknesses:

My feeling is that the concepts of learning and performance are often conflated, and performance differences interpreted as differences in learning.

Statistics are missing to support some of the claims.

Extent of overlap/independence of behavioural measures was not entirely clear to me. E.g. freezing and 'pausing'.

Regarding the differences between learning and performance. It is my view that only really performance on the recall test can be thought of as a true measurement of learning, and that all the differences reported during acquisition might reflect learning differences but might also affect performance differences (or a mixture of both). For example, in my experience, footshock causes mice to have several locomotor responses (e.g. running, jumping etc), and as noted by the authors themselves, animals that take longer to drink chocolate milk delivered to the magazine might accrue more head entry measurements. If male or female mice differ on these measures systematically, it could affect acquisition performance. Moreover, it could do so differentially in single versus mixed valence paradigms. My suggestion is that the authors insert a bit more caution into their interpretations of acquisition performance as reflecting 'learning' differences.

Figures 3-4 have statistics in the caption, but I couldn't find statistics for the data reported in Figures 1 and 2. These should be reported in text or in the caption (in a manner consistent with Figures 3-4).

Regarding behavioural overlap/independence of behavioural measures, as the authors have done such a thorough job of assessing behaviour, and used regressions, I feel like this has probably been taken into account. However, I am not sure this has been communicated as well as it could be. For instance, when a mouse is being measured for pausing, could it also be scored as freezing? Or are the authors thinking of these as separable constructs? If so, what makes them separable? If this could be spelled out a bit more clearly that would be helpful to readers like me.

Figure 1B-E – is the x axis showing Day (as in 1A) or seconds? This is confusing and should be labelled. It reads as Day as it is the same scale as Figure 1A (labelled as Day), but in the text it is mentioned that significant differences in responding are only observed in the last 10 seconds of the CS presentation, which appears to correlate with the significance lines and asterisks.

On page 10 it is suggested that there is a "lack of evidence of acquisition observed with standard behavioural metrics". I found this confusing because I thought that there is evidence of acquisition using these metrics, as reported in Figure 1 (only not for males in the mixed valence learning). That is, the animals do diverge in their head entries/freezing as appropriate in the last 5 secs of CS presentation, which makes perfect sense as this is when the outcomes are delivered. So I'm not entirely sure what the authors mean here and suggest that they make it clearer.

Page 3, I believe there should be a full stop (period) after the brackets containing references?

Page 5 – Is "analysis data" meant to read "data analysis"?

Is mixed-valence paradigm learning superior or just different (page 14)? I would argue that it's just different. As mentioned before, it is only really the test data that can be thought of as a true measure of learning, and I am not sure this claim can be made unless the authors are directly comparing the amount of learning to an appetitive CS or aversive CS across the two paradigms. I don't see these direct comparisons reported so I am not sure this claim can be made. I also feel that, in life, there are paradigms that are primarily aversive or appetitive, so I'm not sure it's even useful to try and claim that one type of learning is superior over another.

Cathy S. Chen is listed as a reference when all the other references just refer to the last name. Suggest consistency E.g. Page 18.

*Reviewer #2 (Recommendations for the authors):*

This manuscript by Schuler and colleagues describes a powerful set of analyses that reveal that our assumptions about how our measurements of affective behaviors, especially pavlovian conditioned responses, may be influenced by sex, as well as species.

A first strength is the use of unbiased pose estimation techniques to understand mouse behavior in these conditioning paradigms, rather than assuming the standard measures are the whole story. The vast majority of behavioral tasks for mice were "ported" from (usually male) rats, and this has sometimes led to belated recognition that we may be failing to account for species-specific affective behaviors in mice. This can lead, for example, to assumptions that mice "can't learn" paradigms, when it may be more accurate to say we are simply not measuring their learning appropriately. It is certainly the case that Figure 1 and Figure S2 present a mediocre picture of learning using some standard measures. However, by using pose estimation and applying a linear classifier to the full set of individual behavioral postures, the authors are able to classify behaviors as being conditioned responses to a reward or shock-associated cue in Figure 2, with some intriguing nuances due to sex that they explore in the rest of the paper. These data are convincing in part because the behavior contributing most to the classifier for reward associated cues are port orientation behaviors, and the behavior contributing to shock associated cues are pausing – in other words, the same classes of valenced behavior previously linked with these behaviors are showing up again, but using techniques that accommodate features of how mice move and behave. We have long known that pavlovian conditioned responses are multimodal, and novel techniques like this may now allow us to analyze all relevant behaviors.

A second strength is the careful analysis of how exploration interacts with the valence of events to lead to sex biases in behavior. They find that exploratory behavior (defined as rearing and locomoting) is negatively correlated in all animals with the strength of conditioning for both shock and reward associated cues – but they also find that male mice are more likely to continue to show these behaviors to a neutral cue during reward conditioning. Conversely, female mice are more likely to engage non-cue elicited pause behaviors by shock experiences. Collectively, these findings suggest that individual biases in exploratory behavior, including those associated with sex, have to be taken into account when analyzing behaviors. Further, the finding that exploratory behavior is generally negatively correlated with the expression of conditioned responses suggests that there are competing demands in the brain to respond to cues or to explore the environment, and that animals can't do both at once. This is a very interesting implication of this paper.

Weaknesses:

One weakness is generalizability of the findings. The authors had to make some judgement calls about their pavlovian conditioning parameters (how strong, how frequent, etc). A full parametric analysis manipulating these settings is not necessarily warranted, but it is worth considering if these effects needed to be in this "sweet spot" to reveal themselves. This could particularly inform the conclusions about the influence of exploration versus exploratory pauses as the mechanism leading to sex differences in behaviors. In other words, if shock intensity, or number of pairings, or some other variable is manipulated, does this elicit higher levels of nonspecific pausing in male mice? Would a more salient appetitive training paradigm reduce male exploratory behavior during relevant cues? It may be the case that the sex differences they observe in aversive and appetitive learning are differentially influenced by these manipulations, which would point to specific mechanisms regulating sex differences in exploratory behavior. This is a minor weakness and could be the focus of follow up work.

A second related weakness is the missed opportunity to use manipulations to discover the origins of these differences. In addition to task manipulations proposed in the previous paragraph, one could imagine a variety of manipulations to tease out the neural mechanisms regulating exploration during learning, either as a sex difference question or as a neural circuit question within sex (since Figures 2 and 3 suggest a good deal of within-sex variability in these traits). This is a minor weakness and could be the focus of follow up work.

The figures are somewhat challenging to read, in part because the differences between symbols (e.g. circle versus diamond) or the colors of CS- versus CSs were not as easy to discriminate as the authors might have hoped.

There is a typo/run on sentence in the second paragraph of the introduction.

*Reviewer #3 (Recommendations for the authors):*

A data-driven approach is used in this study to provide a comprehensive analysis of mouse behaviour in a single vs mixed valence conditioning paradigm. The inclusion of both sexes is a major strength, and the description of the data-driven approach is appreciated and encourages a broader perspective on how to assess mouse behaviour in response to appetitive and aversive conditioning. While the data-driven approach is clearly described, the findings are rather exploratory. The value of this approach is present in the text, but is not convincing in its current state. Other issues relate to what is lacking from the text. Definitions of the behaviours should be presented in an easy to reference manner. It took me a while to locate the definition of pausing. It is still unclear how pausing is different than freezing, other than the timing of when it is measured. How did the authors come to their choices for exploratory behaviours and to what extent have they been shown to be sex-neutral, being less common in one sex than the other.

The authors briefly mentioned some research on female rats displaying darting behaviour and they state that they were unable to measure this in their mice. It is unclear why this is the case. This behaviour has also been detected in mice, including in a study published in the same journal.

Finally, what are the group sizes?

In addition to addressing the comments above, more emphasis should be placed on the value of this approach for the field, including how it can be adopted by others in preclinical studies addressing neuropsychiatric conditions with known sex differences. The writing can also be improved to more explicitly describe how the available findings can better reflect our understanding of sex differences in neuropsychiatric conditions.

---

## [Author Response]

Reviewer #1 (Recommendations for the authors):Strengths:Highly in-depth behavioural assessments, with multiple behaviours measured and compared to assess aversive and appetitive learning.Interesting data investigating sex differences in baseline exploration – something that is often overlooked.The fact that findings highlight the need to take into consideration how tasks are designed when interpreting studies that demonstrate sex differences or, really, learning differences of any kind.Well written overall.Interesting and insightful discussion.

We thank the reviewer for this thoughtful evaluation of both the strengths and the weaknesses of our work. We would like to address the weaknesses raised by the reviewer in more detail.

Weaknesses:My feeling is that the concepts of learning and performance are often conflated, and performance differences interpreted as differences in learning.

We have edited the language surrounding learning and performance for consistency.

Statistics are missing to support some of the claims.

All statistics to support our claims are present, however, as the reviewer elaborated in the recommendations for authors, not all are contained in text or in figure legends, specifically for results shown in Figure 1 and 2. We made this decision in the interest of readability given the large number of comparisons presented in the analyses included in these figures. The comprehensive statistical results are reported in the supplementary file and this is clearly referenced in the figure legends. Statistical significance is indicated in the figures, and additional detailed information can be easily found by the interested reader in the referenced tables within the supplementary file.

Extent of overlap/independence of behavioural measures was not entirely clear to me. E.g. freezing and 'pausing'.

We have clarified the relationship between freezing and pausing more explicitly on p.10 of the manuscript. In brief, these measures are highly correlated, yet because ‘freezing’ has an established definition with clearly defined criteria, we opted to use the term ‘pausing’ instead. We have further elaborated on the description of the data structure in the method section, clarifying the relationships between the predictors for the reader.

Regarding the differences between learning and performance. It is my view that only really performance on the recall test can be thought of as a true measurement of learning, and that all the differences reported during acquisition might reflect learning differences but might also affect performance differences (or a mixture of both). For example, in my experience, footshock causes mice to have several locomotor responses (e.g. running, jumping etc), and as noted by the authors themselves, animals that take longer to drink chocolate milk delivered to the magazine might accrue more head entry measurements. If male or female mice differ on these measures systematically, it could affect acquisition performance. Moreover, it could do so differentially in single versus mixed valence paradigms. My suggestion is that the authors insert a bit more caution into their interpretations of acquisition performance as reflecting 'learning' differences.

We thank the reviewer for noting the inconsistencies in wording in the manuscript. We agree with the reviewer that there is a difference between learning and performance, which is generally overlooked, but our findings suggest is critical. The reviewer is correct in raising the consideration that the presence of the US influences performance. For this reason, in all analyses behavior is evaluated only during the pre-outcome period, and the period of outcome delivery is excluded to avoid conflating response to the cue with the acute effects of outcome delivery. The reviewer makes the valid and important point that experience during the protocols may affect learning rate. We do not intend to claim that learning is better in the mixed-valence protocol, a point further addressed in response to another of the reviewer’s comment below. Rather, we argue that performance in the mixed-valence protocol provides a better approximation of learning that is less biased by underlying sex-differences than afforded by performance readouts in single-valence protocols. We have edited the language to incorporate the reviewer’s feedback and to ensure consistency in the usage of performance and learning.

Figures 3-4 have statistics in the caption, but I couldn't find statistics for the data reported in Figures 1 and 2. These should be reported in text or in the caption (in a manner consistent with Figures 3-4).

While we agree that statistical results should be reported in the text or figure legends where possible, we have opted to include them in a table within the supplementary file for Figure 1. We made this decision given the large number of contrasts performed for each sub figure, the inclusion of which would make the figure legend challenging to follow. Fully reporting all statistical comparisons for Figure 1 would require 15 days * 3 cue type comparisons for each sex and each model (mixed, appetitive) for Figures B and C (i.e., 90 contrasts total) and 6 days * 3 cue type comparisons for each sex and each model (mixed, aversive) for Figures D and E (i.e., 36 contrasts total). We do not believe that listing the test statistic and significance for 126 contrasts will aid the understanding of the analysis more than it will impair following the description of the analyses. For this reason, we compiled this important information in a supplementary table that is referenced in the legend and invite the reader to access this should they wish to evaluate all the test statistics, their significance and the overall model fit.

Similarly, we have decided against including the test statistic and *p*-values for the individual regression coefficients shown in Figure 2D, which would include a list of 3 models * 28 predictors, totaling 84 comparisons. The reviewer’s comment has, however, made us realize that the original manuscript omitted some of these statistics in the supplementary tables. This has now been fixed and the reference to Supplementary File 1d is included in the figure legend. The distributions in Figure 2E are shown exclusively for visualizing the raw data, but not to assess or assert statistical significance.

Regarding behavioural overlap/independence of behavioural measures, as the authors have done such a thorough job of assessing behaviour, and used regressions, I feel like this has probably been taken into account. However, I am not sure this has been communicated as well as it could be. For instance, when a mouse is being measured for pausing, could it also be scored as freezing? Or are the authors thinking of these as separable constructs? If so, what makes them separable? If this could be spelled out a bit more clearly that would be helpful to readers like me.

We have clarified the relationship between freezing and pausing more explicitly now on p.10:

“Pausing is strongly correlated with ‘*freezing’* (Figure 2—figure supplement 3), suggesting that the pausing syllable is capturing at least partially what is classically considered ‘*freezing’*. However, we refrain from labeling the syllable ‘*freezing*, because ‘*freezing’* is conceptualized by explicitly defined criteria, such as absence of all movement for a minimum duration, which we do not probe in this data-driven approach (Blanchard and Blanchard, 1969).”

We also include more extensive reasoning on the use of PLS-DA as a model, and thereby a more detailed summary of the data structure in the methods section as well (p.7). We believe that this clarifies the relationships between predictors to the reader:

“Partial least squares discriminant analysis (PLS-DA). PLS-DA is a supervised statistical approach used for classification and dimensionality reduction (Rosipal and Krämer, 2006). Using syllable frequencies as predictors makes them inherently correlated, because animals can only be assigned one syllable at a time. In addition, the location-based predictors are likely correlated with syllable frequency, as some syllables may be performed preferentially at specific locations within the condition box. PLS-DA overcomes this challenge in the data structure, as it performs well with large numbers of correlated predictors, and inherently provides classification and feature selection.”

Figure 1B-E – is the x axis showing Day (as in 1A) or seconds? This is confusing and should be labelled. It reads as Day as it is the same scale as Figure 1A (labelled as Day), but in the text it is mentioned that significant differences in responding are only observed in the last 10 seconds of the CS presentation, which appears to correlate with the significance lines and asterisks.

We thank the reviewer for bringing this to our attention. We have added axis labels to Figures 1B-E.

On page 10 it is suggested that there is a "lack of evidence of acquisition observed with standard behavioural metrics". I found this confusing because I thought that there is evidence of acquisition using these metrics, as reported in Figure 1 (only not for males in the mixed valence learning). That is, the animals do diverge in their head entries/freezing as appropriate in the last 5 secs of CS presentation, which makes perfect sense as this is when the outcomes are delivered. So I'm not entirely sure what the authors mean here and suggest that they make it clearer.

We have made the following edits to the sentence:

“Overall, this suggests that the lack of evidence of **learning** observed with standard behavioural metrics **on recall day** reflects the limitations of the metrics in a mixed-valence setting rather than a lack of learning.”

These edits also reflect the reviewer’s first comment on performance and learning. We again would like to emphasize that all data used to analyze behavior excludes the outcome delivery period to give a more accurate readout of performance that is not influenced by the acute effect of outcome delivery. This information is included in the figure legend of Figure 1 as well as the main text (p.7).

Page 3, I believe there should be a full stop (period) after the brackets containing references?

We thank the reviewer for bringing this to our attention. The error has been fixed.

Page 5 – Is "analysis data" meant to read "data analysis"?

We thank the reviewer for bringing this to our attention. The error has been fixed.

Is mixed-valence paradigm learning superior or just different (page 14)? I would argue that it's just different. As mentioned before, it is only really the test data that can be thought of as a true measure of learning, and I am not sure this claim can be made unless the authors are directly comparing the amount of learning to an appetitive CS or aversive CS across the two paradigms. I don't see these direct comparisons reported so I am not sure this claim can be made. I also feel that, in life, there are paradigms that are primarily aversive or appetitive, so I'm not sure it's even useful to try and claim that one type of learning is superior over another.

We thank the reviewer for the opportunity to clarify this point further. Specifically, we are not asserting that the learning in the mixed-valence paradigm is superior, but that “Our analyses highlight the superiority of mixed-valence paradigms in minimizing baseline sex differences in exploratory behaviors, thereby allowing more accurate assessment of learning” (p.14)

Through the analyses in the manuscript, we demonstrate that performance in the mixed-valence protocol is less influenced by baseline sex-differences in exploration, thereby making performance readouts a more accurate approximation of learning that is less biased by sex-differences in exploration. In contrast, we show that in the single-valenced protocols, sex-differences are amplified, especially in the case of the aversive-only conditioning in females. We do not claim that the learning is better in one protocol or the other, and as the reviewer noted, we are not attempting to directly compare performance metrics across protocols to claim that there is more or less learning observed.

We hope that clarifying the language around performance and learning, following this reviewer’s suggestion, has helped to clarify this point. We have also rephrased the sentence in the manuscript to avoid this misinterpretation:

“Our analyses demonstrate that performance in the mixed-valence protocol is less biased by baseline sex differences in exploratory behaviors, thereby allowing more accurate assessment of learning.”

Cathy S. Chen is listed as a reference when all the other references just refer to the last name. Suggest consistency E.g. Page 18.

We thank the reviewer for bringing this to our attention. The error has been fixed.

Reviewer #2 (Recommendations for the authors):This manuscript by Schuler and colleagues describes a powerful set of analyses that reveal that our assumptions about how our measurements of affective behaviors, especially pavlovian conditioned responses, may be influenced by sex, as well as species.A first strength is the use of unbiased pose estimation techniques to understand mouse behavior in these conditioning paradigms, rather than assuming the standard measures are the whole story. The vast majority of behavioral tasks for mice were "ported" from (usually male) rats, and this has sometimes led to belated recognition that we may be failing to account for species-specific affective behaviors in mice. This can lead, for example, to assumptions that mice "can't learn" paradigms, when it may be more accurate to say we are simply not measuring their learning appropriately. It is certainly the case that Figure 1 and Figure S2 present a mediocre picture of learning using some standard measures. However, by using pose estimation and applying a linear classifier to the full set of individual behavioral postures, the authors are able to classify behaviors as being conditioned responses to a reward or shock-associated cue in Figure 2, with some intriguing nuances due to sex that they explore in the rest of the paper. These data are convincing in part because the behavior contributing most to the classifier for reward associated cues are port orientation behaviors, and the behavior contributing to shock associated cues are pausing – in other words, the same classes of valenced behavior previously linked with these behaviors are showing up again, but using techniques that accommodate features of how mice move and behave. We have long known that pavlovian conditioned responses are multimodal, and novel techniques like this may now allow us to analyze all relevant behaviors.A second strength is the careful analysis of how exploration interacts with the valence of events to lead to sex biases in behavior. They find that exploratory behavior (defined as rearing and locomoting) is negatively correlated in all animals with the strength of conditioning for both shock and reward associated cues – but they also find that male mice are more likely to continue to show these behaviors to a neutral cue during reward conditioning. Conversely, female mice are more likely to engage non-cue elicited pause behaviors by shock experiences. Collectively, these findings suggest that individual biases in exploratory behavior, including those associated with sex, have to be taken into account when analyzing behaviors. Further, the finding that exploratory behavior is generally negatively correlated with the expression of conditioned responses suggests that there are competing demands in the brain to respond to cues or to explore the environment, and that animals can't do both at once. This is a very interesting implication of this paper.Weaknesses:One weakness is generalizability of the findings. The authors had to make some judgement calls about their pavlovian conditioning parameters (how strong, how frequent, etc). A full parametric analysis manipulating these settings is not necessarily warranted, but it is worth considering if these effects needed to be in this "sweet spot" to reveal themselves. This could particularly inform the conclusions about the influence of exploration versus exploratory pauses as the mechanism leading to sex differences in behaviors. In other words, if shock intensity, or number of pairings, or some other variable is manipulated, does this elicit higher levels of nonspecific pausing in male mice? Would a more salient appetitive training paradigm reduce male exploratory behavior during relevant cues? It may be the case that the sex differences they observe in aversive and appetitive learning are differentially influenced by these manipulations, which would point to specific mechanisms regulating sex differences in exploratory behavior. This is a minor weakness and could be the focus of follow up work.A second related weakness is the missed opportunity to use manipulations to discover the origins of these differences. In addition to task manipulations proposed in the previous paragraph, one could imagine a variety of manipulations to tease out the neural mechanisms regulating exploration during learning, either as a sex difference question or as a neural circuit question within sex (since Figures 2 and 3 suggest a good deal of within-sex variability in these traits). This is a minor weakness and could be the focus of follow up work.

We thank the reviewer of their detailed evaluation of both the strengths and the weaknesses of our work. We agree that the parameter space in the current work is restricted, and that our findings suggest that stimulus salience will be a fruitful avenue to explore in future work. In the discussion, we elaborate on the limitations of the present experiments with regards to the generalizability of the chosen stimuli similar to the reviewer’s argument. In addition, we agree that the sex differences in behavior in the single-valence protocols posit an interesting target for follow-up work to identify their underlying biological mechanisms and have incorporated this in the discussion.

The figures are somewhat challenging to read, in part because the differences between symbols (e.g. circle versus diamond) or the colors of CS- versus CSs were not as easy to discriminate as the authors might have hoped.

We appreciate the reviewer’s observation. Throughout the manuscript we have aimed for consistent visualization of the three main factors in our study: Protocol (Shape), CS Type (Color), and Sex (Fill). Due to the complexity of the design, we have limited opportunity to further distinguish the different levels we are showing through visual means. Where possible, we have additionally added annotations to guide the viewer (e.g., using headers as in Figure1B-E).

We agree that the distinction between the circle and diamond shape is particularly hard to notice. For this reason, we have altered the diamond shape to be more distinct from the circle to improve the legibility of the figures. With regards to the colors, we chose the color palette to be color-blind friendly (Viridis palette) and selected three colors along this palette with large distances that contrast well against a white background. Changing colors may facilitate distinctions for many viewers but would likely compromise colour-blind accessibility. We therefore chose not to alter the chosen color theme.

There is a typo/run on sentence in the second paragraph of the introduction.

We thank the reviewer for bringing this to our attention. The error has been fixed.

Reviewer #3 (Recommendations for the authors):A data-driven approach is used in this study to provide a comprehensive analysis of mouse behaviour in a single vs mixed valence conditioning paradigm. The inclusion of both sexes is a major strength, and the description of the data-driven approach is appreciated and encourages a broader perspective on how to assess mouse behaviour in response to appetitive and aversive conditioning. While the data-driven approach is clearly described, the findings are rather exploratory. The value of this approach is present in the text, but is not convincing in its current state.

We thank the reviewer for the opportunity to clarify the value of the exploratory nature of our approach. Importantly, we wish to underscore that we view the unbiased, data-driven approach taken in the manuscript as a strength rather than a weakness, which defines the rigor and validity of the current findings. In the provided review, the specific reasons why the reviewer qualifies the current state as not convincing are not explained.

We would like to emphasize, that we employed this data-driven approach explicitly to support a rigorous exploratory analysis, in that we our goal was to identify the best predictors of cue discrimination in a novel task without the inherent bias introduced by relying on pre-conceived notions about what the behavior should be. Notably, when we applied existing metrics, we fail to see evidence of reward learning in both sexes. Applying a fully hypothesis-driven approach (i.e., freezing = fear learning, head entries = reward learning), would have missed many of the important nuances reported throughout the manuscript, and led to the erroneous conclusion that mice fail to acquire the reward associations.

We thoughtfully selected methods to identify predictive metrics without compromising statistical rigour. For example, we could have chosen to compare the frequency of each syllable during the different cue types in each model using standard ANOVAs, however, this would ignore inherent collinearity between the syllables and would likely inflate Type-I errors unless appropriately corrected for, which this design is not suited for with the given sample size. These limitations are optimally addressed by our chosen method PLS-DA which provides a statistically rigorous means to identify predictive metrics. In addition to using a robust statistical method for discovering metrics, we rigorously validated these findings in test datasets that are entirely independent from the training cohort that we used to build the classifier. This additional step is important in ensuring that our findings not only are statistically rigorous but also reproducible across cohorts.

To address this comment, we have expanded the discussion to more explicitly highlight the strength of our approach:

“In line with previous research, our findings also emphasize the power of rigorously applied data-driven approaches in behavioral neuroscience in re-evaluating long-standing assumptions and identifying sex-specific phenotypes in behaviors previously established exclusively on male data (Levy et al., 2023; Schuler et al., 2024; Shansky, 2024). Our data show that relying exclusively on existing behavioral metrics can be misleading when developing a novel behavioral protocol. Data-driven approaches, as the one employed in this study, remove biased and preconceived notions from the interpretation of animal behavior, leading to identification of behaviors solely revealed from the data structure with minimal human interference. PLS-DA provides a novel way to understand and interpret the high dimensional data produced by pose estimation and behavioral classifiers. Behavioral classifiers can identify a large number of syllables, and rigorously comparing the duration spent in each syllable between outcomes of interest is not feasible due to the large amount of multiple testing corrections required in combination with usually small sample sizes in behavioral neuroscience experiments. As compared to unsupervised dimensionality reduction methods, such as principal component analysis, PLS-DA identifies axes of variance in the predictors that are correlated with the outcome. This makes it a useful approach in that it explicitly captures features that play an important role in outcome prediction. We present a blueprint for rigorously identifying relevant behavioral metrics of learning in both sexes that can be applied across a wide variety of paradigms, including animal models of neuropsychiatric disease.”

In addition, and in line with suggestions from Reviewer 1, we have further expanded on the choice of PLS-DA as a statistical approach and why it is well-suited for the given dataset as follows:

“Partial least squares discriminant analysis (PLS-DA). PLS-DA is a supervised statistical approach used for classification and dimensionality reduction (Rosipal and Krämer, 2006). Using syllable frequencies as predictors makes them inherently correlated, because animals can only be assigned one syllable at a time. In addition, the location-based predictors are likely correlated with syllable frequency, as some syllables may be performed preferentially at specific locations within the condition box. PLS-DA overcomes this challenge in the data structure, as it performs well with large numbers of correlated predictors, and inherently provides classification and feature selection.”

Other issues relate to what is lacking from the text. Definitions of the behaviours should be presented in an easy to reference manner. It took me a while to locate the definition of pausing. It is still unclear how pausing is different than freezing, other than the timing of when it is measured. How did the authors come to their choices for exploratory behaviours and to what extent have they been shown to be sex-neutral, being less common in one sex than the other.

We thank the reviewer for highlighting these opportunities to improve the manuscript. We have made the following changes to clarify the specific points:

To easily reference individual syllables, we have added a supplementary figure showing the trajectory plots of the individual syllables in Figure 2—figure supplement 1. We have further uploaded grid movies showing examples of the individual syllables. This information was used in combination by the experimenter to label the syllables, and the trajectory plots by themselves can sometimes be misleading (e.g., pausing and licking look the same when considering the trajectory plots, but licking is clearly seen to occur in the food port when evaluating the videos). In addition, we have elaborated on how syllables were assigned to the individual categories where we felt an additional explanation is needed (e.g., attending). This is further described in the figure legend to Figure 2—figure supplement 1 as well.

We have added a supplementary figure showing the correlation between freezing and pausing, and we added the following sentence on p.10 to better explain our reasoning for choosing to treat pausing conceptually different from freezing:

“Pausing is strongly correlated with ‘*freezing’* (Figure 2—figure supplement 3), suggesting that the pausing syllable is capturing at least partially what is classically considered ‘*freezing’*. However, we refrain from labeling the syllable ‘*freezing*, because ‘*freezing’* is conceptualized by explicitly defined criteria, such as absence of all movement for a minimum duration, which we do not probe in this data-driven approach (Blanchard and Blanchard, 1969).”

We direct the reviewer to the following section from p.10 of the manuscript outlining the observations that led us to hypothesise that sex differences may be explained by differences in exploration:

“Intriguingly, predicting CS^S^ from CS^-^ revealed distinct effects of sex. Females rapidly discriminate CS^S^ from CS^-^ in the mixed paradigm but, in the aversive-only paradigm, the CS^-^ is consistently misclassified as CS^S^ throughout training, with discrimination only evident at recall (Figure 3A,B). In contrast, males similarly discriminate CS^S^ from CS^-^ in both paradigms. The misclassification of CS^R^ as CS^-^ in males and CS^-^ as CS^S^ in females suggests that males are exhibiting more CS^-^ associated behaviours (‘*rearing’*, ‘*locomoting’*) during the CS^R^ and that females are showing less of these same behaviours during the CS^-^ and more of the CS^S^ associated *pausing* behaviours. Integrating these observations led us to hypothesize that a common underlying factor of exploration might explain the observed sex differences in cue responding in single valence protocols.”

Our choice to call these behaviors ‘exploratory’ is in line with the ‘search’ category of the Stanford Mouse Behavior Ethogram, listing locomotion and rearing as non-stimuli driven, general exploratory behaviors (https://med.stanford.edu/mousebehavior/ethogram/active-behaviors/general-activity/exploratory-behavior/search.html).

Whether these behaviors are ‘sex-neutral’ is addressed in the manuscript on p. 11 and shown in Figure 3C. In fact, we demonstrate that the amount of time males and females explore differs at baseline and therefore would not be considered sex-neutral. If the reviewer is interested in the individual usage of the individual syllables summarised in exploration behaviors, we refer them to Figure 2—figure supplement 4.

The authors briefly mentioned some research on female rats displaying darting behaviour and they state that they were unable to measure this in their mice. It is unclear why this is the case. This behaviour has also been detected in mice, including in a study published in the same journal.

We thank the reviewer for the opportunity to clarify this apparent contradiction with previous research. First, we would like to re-emphasize that darting as a sex-specific response to fear conditioning has been primarily reported in rats (Gruene et al., 2015; Mitchell et al., 2024, 2022). While some research has found darting to be present in mice as well, this has not been reported as a sex-specific response in female mice and has been primarily investigated in male mice (Borkar et al., 2024, 2020; Fadok et al., 2017; Dong et al., 2019). In addition, the validity of a flight-like response to conditioned cues in mice as reported in previous literature has been challenged. That is, it has been proposed that flight-like responses in mice are potentially an innate response driven by characteristics of the conditioned stimulus when the conditioned stimulus is white noise (Trott et al., 2022). We have added more references to the relevant Discussion section to address this comment. Finally, darting has not been previously reported in a mixed-valence context in rats.

We would also like to clarify that we were not unable to measure darting in our mice but rather it did not emerge as a predictor. We would hypothesize that darting would show up in our syllables, likely as a type of behavior in the ‘Jump’ or ‘Locomotion’ group, or a combination thereof. If darting were a predictor of the CS^S^ in our protocols, PLS-DA would have identified it as such. However, the primary evidence for CS^S^ association is observed in ‘pausing’, with no positively predictive value of other movement syllables, as shown in Figure 2D. We have altered the wording in the discussion to clarify this point to the readers.

Finally, what are the group sizes?

The samples sizes for each experimental cohort are reported in the ‘Materials and methods’ section, under ‘Classical Conditioning Protocols’ / ‘Conditioning cohorts’ (p.5) and individual mice are visualized as data-points throughout the manuscript:

“Conditioning cohorts. The mixed-valence discovery and validation cohort consisted of 56 (n*_Male_* = 28, n_*Female*_ = 28) and 18 (n*_Male_* = 9, n_*Female*_ = 9) mice, respectively. The single-valence cohorts consisted of 16 animals each (n*_Male_* = 8, n_*Female*_ = 8).”

In addition to addressing the comments above, more emphasis should be placed on the value of this approach for the field, including how it can be adopted by others in preclinical studies addressing neuropsychiatric conditions with known sex differences. The writing can also be improved to more explicitly describe how the available findings can better reflect our understanding of sex differences in neuropsychiatric conditions.

We have taken this comment into consideration, and this is further addressed above